# Current Understanding of the Immunomodulatory Activities of High-Density Lipoproteins

**DOI:** 10.3390/biomedicines9060587

**Published:** 2021-05-21

**Authors:** Athina Trakaki, Gunther Marsche

**Affiliations:** Division of Pharmacology, Otto Loewi Research Center, Medical University of Graz, Universitätsplatz 4, 8010 Graz, Austria; athina.trakaki@medunigraz.at

**Keywords:** high-density lipoprotein, HDL function, immunomodulation, neutrophils, monocytes, macrophages, dendritic cells, T cells, eosinophils

## Abstract

Lipoproteins interact with immune cells, macrophages and endothelial cells - key players of the innate and adaptive immune system. High-density lipoprotein (HDL) particles seem to have evolved as part of the innate immune system since certain HDL subspecies contain combinations of apolipoproteins with immune regulatory functions. HDL is enriched in anti-inflammatory lipids, such as sphingosine-1-phosphate and certain saturated lysophospholipids. HDL reduces inflammation and protects against infection by modulating immune cell function, vasodilation and endothelial barrier function. HDL suppresses immune cell activation at least in part by modulating the cholesterol content in cholesterol/sphingolipid-rich membrane domains (lipid rafts), which play a critical role in the compartmentalization of signaling pathways. Acute infections, inflammation or autoimmune diseases lower HDL cholesterol levels and significantly alter HDL metabolism, composition and function. Such alterations could have a major impact on disease progression and may affect the risk for infections and cardiovascular disease. This review article aims to provide a comprehensive overview of the immune cell modulatory activities of HDL. We focus on newly discovered activities of HDL-associated apolipoproteins, enzymes, lipids, and HDL mimetic peptides.

## 1. Introduction

From an evolutionary point of view, lipoproteins are not only described as lipid transporters, but are also known to display important immunomodulating functions. Specifically, of all lipoproteins, high-density lipoprotein (HDL) particles have the highest affinity for binding and neutralizing pathogen-associated lipids, such as lipoteichoic acid and lipopolysaccharide (LPS) [1,2], which are responsible for mediating excessive immune activation during bacterial infections [1,3,4]. This is thought to be of considerable importance in septic conditions [5], reflected by an inverse association of HDL cholesterol with death from infection [6] along with sepsis severity and morbidity [1]. Moreover, infusion of the apolipoprotein (apo) A-I mimetic peptide 4F decreased mortality and morbidity in experimental sepsis models [7]. In addition, HDL inhibits endothelial cell adhesion molecules, including vascular cell adhesion molecule 1 (VCAM-1), intercellular adhesion molecule-1 (ICAM-1) and E-selectin, which are responsible for the binding of monocytes at sites of developing atherosclerosis [8]. Interestingly, HDL is also reported to have anti-parasitic effects [1].

However, an increasing number of studies have shown that chronic systemic inflammatory disorders significantly affect HDL composition and function. Such disorders include systemic lupus erythematosus and rheumatoid arthritis [9,10,11,12], atrial fibrillation [13], psoriasis [14,15,16,17,18,19,20,21,22], chronic kidney disease [23,24,25,26], liver failure [27], as well as allergic and skin diseases, including allergic rhinitis [28,29,30], asthma [31,32,33] and atopic dermatitis [34]. In turn, altered HDL function may have a significant impact on the progression of the disease and influence the risk of infections and cardiovascular disease [35].

This review provides a comprehensive overview of immune cell modulatory activities of HDL and its associated proteins, lipids and enzymes. We focus on newly discovered effects of HDL-associated apolipoproteins (and mimetic peptides of apolipoproteins), HDL-associated enzymes, such as paraoxonase (PON), as well as HDL-associated lipids, including sphingosine-1-phosphate (S1P) and lysophospholipids, on the function of immune cells; specifically dendritic cells, monocytes, macrophages, neutrophils, eosinophils and T cells.

## 2. HDL Composition

### 2.1. HDL-Associated Apolipoproteins and Lipids

HDL particles consist of an outer amphipathic layer of several apolipoproteins, including apoA-I, apoA-II, apoA-IV, apoCs, apoD, apoJ and apoM and many other proteins enriched in lesser amounts [36]. Besides apolipoproteins, free cholesterol, phospholipids (mainly glycerophospholipids), lysophospholipids [37,38] such as lysophosphatidylcholine (LPC) species [39], and sphingolipids, such as sphingomyelin and S1P [40,41,42], are found on the surface of HDL particles. At the same time, the hydrophobic core consists of cholesteryl esters and triglycerides [37].

### 2.2. HDL-Associated Enzymes

HDL particles carry important enzymes, such as paraoxonase, which can exert a protective effect against oxidative damage of circulating lipoproteins and cells [43]. Among the enzymes of the paraoxonase family, paraoxonase 1 (PON1) and paraoxonase 3 (PON3) are mainly associated with HDL and exhibit paraoxonase, lactonase and esterase activities [44]. PON1 protects HDL and low-density lipoprotein (LDL) against oxidation, reduces macrophage and aortic oxidative status associated with decreased superoxide anion production, and enhances the reverse cholesterol transport by facilitating the binding of HDL particles to macrophages [45,46,47,48]. Other HDL-associated enzymes are platelet-activating factor acetylhydrolase (PAF-AH), also known as lipoprotein-associated phospholipase A2 (Lp-PLA2), which is the major enzyme catabolizing potent inflammation mediators, such as platelet-activating factor (PAF) and PAF-like lipids [49,50,51]. Lecithin-cholesterol acyltransferase (LCAT), which is responsible for free cholesterol to cholesteryl ester esterification [52], and phospholipid transfer protein (PLTP), a lipid transfer protein implicated in the remodeling of HDL particles [35,53], are important HDL-associated enzymes.

## 3. Immunomodulatory Functions of HDL

HDL seems to have evolved as part of the innate immune system since the HDL proteome consists of apolipoproteins involved in lipid metabolism and many immune regulatory proteins [37]. In addition to this, HDL-associated lipids, such as S1P and lysophosphatidylcholine, are potent immune modulators.

### 3.1. HDL-Associated Lipids Show Potent Immunomodulatory Functions

S1P is a potent bioactive sphingolipid generated by sphingosine phosphorylation via the sphingosine kinase (SPHK) [54]. Approximately 65–80% of S1P in plasma is associated with HDL, mainly bound to apoM. S1P is rapidly degraded by intracellular S1P lyase or dephosphorylated by S1P phosphatases in most cells; however, its levels in the blood and lymph are in the range of nanomolar to micromolar [55]. HDL-associated S1P is less susceptible to degradation than free S1P or S1P bound to albumin, suggesting an important role of HDL in regulating the uptake, systemic function and cellular degradation of S1P. Although the mechanism of S1P efflux from cells to HDL is not clearly established, it involves specific transporters, such as adenosine triphosphate (ATP)-binding cassette family transporters [56,57,58], including ATP-binding cassette subfamily A member 1 (ABCA1) [59,60,61]. S1P plays a central protective role in the pathogenesis of many inflammatory disorders, including asthma, rheumatoid arthritis and atherosclerosis, through modulation of endothelial barrier function [62,63,64,65] and macrophage function [66]. In addition to this, S1P is associated with signals for immune cell activation and differentiation, such as calcium mobilization, chemotaxis and cytoskeletal reorganization [67].

Circulating lysophosphatidylcholines are carried by HDL and are intensively studied in the context of inflammation. Their concentration can increase dramatically in inflammatory states [68,69]. Lysophosphatidylcholines are widely regarded as pro-inflammatory and harmful mediators. Still, an increasing number of recent studies demonstrated potent anti-inflammatory and anti-allergic properties [69,70,71,72]. Lysophosphatidylcholines should be recognized as important homeostatic mediators involved in all stages of vascular inflammation.

Thus, HDL composition and function both in humans and in animal models are associated with altered immune responses. Activation of various cell types is implicated in cell-mediated immunity. The available literature indicates the ability of HDL to affect functions of dendritic cells, monocytes, macrophages, and lymphocytes. This occurs mainly through the modulation of cholesterol content in lipid rafts, which strongly influences immune cell activation [73].

### 3.2. HDL and Monocyte Function

#### 3.2.1. Effects of HDL and HDL-Associated Apolipoproteins and Lipids on Monocyte Function

Monocytes are heterogeneous cells that circulate in the blood and play a crucial role in innate immunity. During inflammation, monocytes circulate through the blood and extravasate into inflamed tissues, providing nonspecific protection against foreign pathogens, mainly through mechanisms such as phagocytosis and cytokine production [73].

HDL and apoA-I were shown to suppress the expression of the adhesion molecule cluster of differentiation (CD) 11b of monocytes and monocyte adhesion to endothelial cells [74]. Moreover, HDL reduces monocyte inflammatory response in humans [74], while both HDL and apoA-I inhibit macrophage colony-stimulating factor (M-CSF)-induced monocyte spreading through the decrease of cell division control protein 42 homolog (Cdc42) levels [75]. These data suggested that HDL prevents monocyte cytoskeletal reorganization, a step required for migration towards a chemotactic signal [76]. However, it has been suggested that apoA-II enhances the response of monocytes to LPS by suppressing the inhibitory activity of LPS-binding protein. This suggests a pro-inflammatory function of apoA-II in controlling the host response to bacterial LPS and raises the possibility that apoA-II plays a role in antimicrobial host defense [77].

In addition, apoC-III was recently identified to activate the nod-like receptor family pyrin domain-containing 3 (NLRP3) inflammasome in human monocytes by inducing an alternative NLRP3 inflammasome via caspase-8 and dimerization of toll-like receptor (TLR) 2 and TLR4. This suggests that apoC-III inhibition might comprise a potential therapeutic target for vascular and kidney diseases [78]. ApoC-III increased monocyte adhesion to endothelial cells under static and flow conditions [79] and expression of VCAM-1 and ICAM-1 on endothelial cells, via activation of protein kinase C-β and nuclear factor-κΒ (NF-κB) [80]. It has been suggested that low HDL cholesterol levels, modified/dysfunctional apoA-I and reduced expression of ABCA1/ATP-binding cassette subfamily G member 1 (ABCG1) in monocytes/macrophages may be sufficient to induce inflammasome activation in humans [81]. Such changes occur commonly in patients with chronic kidney disease, poorly controlled type 2 diabetes and aging [82,83,84,85,86,87,88]. Interestingly, HDL isolated from allergic rhinitis patients and psoriasis patients under treatment with biologics also showed significantly impaired capacity to suppress NF-κB and subsequent pro-inflammatory cytokine secretion in a human monocyte cell line [28,89].

Approximately 5% of all HDL particles contain apoM. ApoM-containing HDL was shown to inhibit Cu^2+^-induced LDL oxidation and to stimulate cholesterol efflux from THP-1 foam cells more efficiently than apoM-lacking HDL [90]. Moreover, apoM-containing HDL or recombinant apoM-bound S1P reduced endothelial cell adhesion to monocytes by reducing the abundance of adhesion molecules VCAM-1 and E-selectin, but not ICAM-1, and maintained endothelial barrier integrity. In contrast, apoM alone and apoM-lacking HDL induced opposite effects [91]. The activation of the S1P receptor 1 was sufficient and essential to promote this anti-inflammatory effect [91].

HDL inhibited monocyte adhesion and spreading on endothelial cells under shear-flow conditions and suppressed migration in response to the chemokine monocyte chemoattractant protein-1 (MCP-1) [76]. The capacity of HDL from healthy subjects to inhibit MCP-1 production, reactive oxygen species generation and nicotinamide adenine dinucleotide phosphate (NADPH) oxidase activation appeared to be mediated by S1P and sphingosylphosphorylcholine, two lysosphingolipids present on HDL [92].

#### 3.2.2. Effects of HDL-Associated Enzymes on Monocyte Function

Differentiation of monocytes into macrophages and the subsequent process of foam cell formation is the first stage of atherosclerosis development [93]. HDL-associated PON1 was shown to inhibit monocyte-to-macrophage differentiation via inhibition of CD11b and CD36 expression and of total cellular peroxides during phorbol-12-myristate-13-acetate-induced THP-1 monocytes differentiation [94]. The authors of the respective study concluded that this effect could be related to PON1 peroxidase-like activity [94]. PON1 is known to hydrolyze the pro-inflammatory mediator platelet-activating factor, which activates monocytes and leads to their transformation into macrophages [95]. Another study demonstrated that PON1 could reduce monocyte chemotaxis and adhesion to endothelial cells, resulting from the oxidation of palmitoyl, linoleoyl glycerophosphorylcholine [96]. Moreover, in vitro studies showed that HDL-associated paraoxonase and PAF-AH potently inhibit monocyte transmigration in response to oxidized LDL [97]. This ability was reduced in acute inflammatory states due to the accumulation of serum amyloid A (SAA) in HDL particles [98]. PON3 was also shown to inhibit monocyte activation and LDL oxidation [99,100,101]. An older study showed that HDL isolated from apoA-II transgenic mice stimulated lipid hydroperoxide formation in arterial wall cells and induced transmigration of monocytes, which was linked to decreased levels of paraoxonase [102]. Upon glycation, both HDL and paraoxonase lost their ability to inhibit monocyte adhesion to human aortic endothelial cells in response to oxidized-LDL in vitro. This fact could potentially contribute to the atherosclerosis acceleration observed in type 2 diabetes patients [103].

#### 3.2.3. Effects of Reconstituted HDL, Mimetic Peptides and Overexpression of ApoA-I/HDL on Monocyte Function

Reconstituted HDL infusion in type 2 diabetes mellitus patients resulted in a reduction of CD11b expression [104]. The overexpression of apoA-I/HDL in diabetic mice improved cholesterol efflux from bone marrow progenitors, suppressed their proliferation, monocyte production and the general recruitability of monocytes into plaques and inflammatory sites and promoted plaque macrophage polarization to the M2, atherosclerosis-resolving state [105].

Moreover, the apoE mimetic peptide Ac-hE18A-NH2 reduced monocyte adhesion in human umbilical vein endothelial cells and interleukin (IL)-6 and MCP-1 secretion and inhibited LPS-induced VCAM-1 expression [106].

The effects of apoB-depleted serum, isolated HDL, reconstituted HDL, HDL-associated apolipoproteins, and mimetic peptides on monocyte function, both in human studies and in studies utilizing animal models are summarized in Table 1.

### 3.3. HDL and Macrophage Function

Monocytes can differentiate into two different types of macrophages upon different cytokine activation. M1 macrophages are regarded as pro-inflammatory and are induced by T helper type 1 (Th1) cytokines, including interferon γ, tumor necrosis factor (TNF) α (TNF-α), IL-2, and LPS. M2 macrophages are implicated in the resolution of inflammation via suppression of cytokine secretion and promotion of wound healing and tissue remodeling [44,108]. Several humoral factors may modify the balance between M1 and M2 phenotypes. Specifically, HDL increased the expression of M2 macrophage markers in mice, resulting in atherosclerotic plaque regression and in changes both in the content and in characteristics of monocyte-derived macrophages [109]. In humans, apoA-I promoted M2 polarization [107]. At the same time, mature HDL appeared not to influence the alternative differentiation of primary human macrophages towards the M2 phenotype [110].

#### 3.3.1. Effects of HDL and HDL-Associated Apolipoproteins on Macrophage Function

It is known that the interaction of HDL with macrophages leads to many cellular responses important for the control of atherosclerosis, such as cholesterol efflux, suppression of TLR4 signaling, reduction of apoptosis during efferocytosis, and modulation of membrane lipid levels to support macrophage migration [76]. HDL and reconstituted HDL were demonstrated to reduce the inflammatory response mediated by TLRs by activating transcription factor 3 [111]. ApoA-I was shown to inhibit TLR2 receptor expression and to decrease NF-κB activation and pro-inflammatory cytokine production in human monocyte-derived macrophages [112]. Another study examined the effect of HDL on macrophage inflammatory response inhibition to the TLR4 ligand LPS [113]. It was observed that the TIR-domain-containing adapter-inducing interferon-β (TRIF)-related adaptor molecule (TRAM)/TRIF arm of the TLR4 signaling branch was significantly suppressed by HDL, suggesting that HDL inhibits both the MyD88 and the TRAM/TRIF actions of TLR4 activation [113]. However, a recent study showed overt pro-inflammatory effects of HDL-mediated passive cholesterol depletion and lipid raft disruption in murine and human primary macrophages in vitro [114]. These pro-inflammatory effects were confirmed in vivo in peritoneal macrophages from apoA-I transgenic mice, which have elevated HDL levels [114]. Several other studies have shown that HDL can bind, sequester and neutralize LPS, thus preventing the activation of monocytes and macrophages [115,116,117]. Specifically, LPS bound to soluble CD14 can be shuttled to HDL and neutralized in a process implicating lipopolysaccharide-binding protein [115,116]. At the same time, HDL can also neutralize LPS by promoting its release from the surface of macrophages and monocytes [117]. Non-insulin-dependent diabetic subjects with cardiovascular disease depicted increased CD14 levels on the surface of CD14++/CD16- monocytes [118]. CD14 is essential for MyD88-independent LPS signaling via TLR4 [119], and it was shown that both HDL and apoA-I can attenuate the monocyte surface expression of CD14 [107,120]. Moreover, HDL and apoA-I inhibited NADPH oxidase activity, p47phox translocation from the cytoplasm to the plasma membrane, and NADPH oxidase 2 expression in human macrophages incubated under high glucose [121]. Furthermore, apoA-I, through ABCA1-dependent cholesterol efflux, suppressed pro-inflammatory signaling of CD40 in macrophages by preventing TNF receptor-associated factor 6 translocation to lipid rafts [122]. Although apoA-II was less effective than apoA-I in cholesterol efflux from macrophages and impaired the effect of apoA-I only when the relative amount of apoA-I to apoA-II was low [123]; interestingly, a recent study demonstrated that the presence of apoA-II in HDL particles enhanced the ABCA1-mediated efflux compared to HDL particles containing apoA-I and no apoA-II [124].

Moreover, apoA-I binding to ABCA1 in macrophages promoted signaling via Janus kinase 2 (JAK2)/signal transducer and activator of transcription 3 (STAT3) pathway [125], suppressing LPS-induced pro-inflammatory cytokines release [76]. Specifically, the interaction of apoA-I with ABCA1 increases phosphorylation, thereafter activating JAK2, which, in turn, increases the binding activity of apoA-I and ABCA1 transporter [126,127,128]. In addition, JAK2 increases the transporter activity of ABCA1 [129,130], an activity known to have an anti-inflammatory effect. Once JAK2 is activated, it then activates STAT3 [125,128], which is independent of the ABCA1 lipid transport activity [131]. ABCA1 contains two potential docking units with STAT3, which are necessary for STAT3 phosphorylation by apoA-I/ABCA1/JAK2 [130]. It has been proposed that the transcription factor STAT3 performs an anti-inflammatory function in macrophages [125,132] and mediates IL-6 signaling pathways [125,128], suggesting that ABCA1 functions as a direct anti-inflammatory receptor owing to JAK2/STAT3 activation [125,131]. However, it has been reported that the JAK2/STAT3 pathway can also exhibit a pro-inflammatory effect [125,128,132,133,134], highlighting the complexity of parallel processes, which require further investigation. STAT3 regulates several fundamental cellular processes, such as cell migration, proliferation, differentiation and inflammation [135]. At the same time, it also regulates apoptosis via induction of apoptosis inhibitor B-cell lymphoma 2 expression [136,137]. Moreover, mutations in ABCA1 violating the ABCA1/STAT3 complex did not affect the ABCA1-mediated cholesterol efflux. However, they abrogated the ability of ABCA1 to suppress cytokine secretion in response to LPS [138]. It has been demonstrated that macrophage cholesterol load associated with ABCA1 inhibition increases IL-6 production [139]. IL-6 controls inflammatory responses associated with the involvement of innate and adaptive immunity [140]. Specifically, IL-6 was reported to reduce the pro-inflammatory response of human macrophages via induction of IL-4 and IL-10 anti-inflammatory cytokines and reduction of IL-1β pro-inflammatory cytokine secretion [139]. Induction of IL-10 by IL-6 may be involved thereafter in activation support of STAT3 in macrophages with the contribution of the specific receptor IL-10R [139,141]. Moreover, IL-6 was reported to induce the ABCA1 expression and to enhance the transporter-mediated cholesterol efflux to apoA-I with the participation of the JAK2/STAT3 pathway [139]. Thus IL-6 production by lipid-loaded macrophages promotes ABCA1 gene expression, which leads to increased ABCA1-mediated cholesterol efflux via JAK2/STAT3 activation, thereby reducing foam cell formation and free cholesterol accumulation [142]. Apart from IL-6, other cytokines can also modulate ABCA1 expression, including interferon γ (IFN-γ), platelet-derived growth factor and IL-1β, which have an inhibitory effect, whereas IL-10 and transforming growth factor-beta 1 have an inducing effect [131]. This suggests that JAK2/STAT3 may represent an important signaling pathway to reduce pro-inflammatory response and accumulation of cellular lipids [128,139,143], which correlates with the anti-inflammatory mechanism of ApoA-I/ABCA1 interaction and activation of the JAK2/STAT3 signaling pathway [125,143].

Overexpression of apoA-I in mice had a protective role against atherosclerosis, in line with promoting macrophage-specific reverse cholesterol transport in vivo [144]. Moreover, HDL derived from human apoA-II transgenic rabbits exerted stronger cholesterol efflux capacity and inhibitory effects on the inflammatory cytokine expression by macrophages in vitro than HDL derived from non-transgenic rabbits [145]. Transgenic rabbits had reduced aortic and coronary atherosclerosis and reduced macrophages in atherosclerotic lesions, suggesting that enrichment of apoA-II in HDL particles has atheroprotective effects and that apoA-II may become a target for treating atherosclerosis [145]. In human apoA-II transgenic mice on a chow diet, overexpression of human apoA-II maintained effective reverse cholesterol transport from macrophages to liver and feces, even in a situation of HDL deficiency [146]. Another study in human apoA-II transgenic mice also indicated an increased ability of plasma of these mice to extract cholesterol from macrophages, implying a potential antiatherogenic effect [147]. Both HDL and apoA-I removed cholesterol from lipid rafts via ABCA1, scavenger receptor class B type 1 (SR-BI) and ABCG1, reducing the inflammatory response in macrophages and inhibiting the ability of antigen-presenting cells to stimulate T-lymphocytes [74,148]. In the presence of an acidic pH, a characteristic of inflammatory tissue sites and human atherosclerotic lesions, HDL particles undergo spontaneous remodeling. An acidic pH promoted forming of lipid-poor apoA-I and the fusion of larger HDL particles [149], enhancing the ability to promote cholesterol efflux from cultured human macrophage foam cells [149]. However, it must be noted that the cholesterol efflux capacity of HDL derived from patients suffering from atrial fibrillation, acute coronary syndrome, chronic kidney disease or psoriasis, was significantly impaired compared to control subjects, as was evaluated in J774 and RAW 264.7 macrophages, respectively [13,14,23,89,150,151,152].

In asymptomatic familial hypercholesterolemia patients, higher macrophage cholesterol efflux capacity, as well as higher S1P and apoM content of HDL, were found, suggesting a potential protective role against premature coronary heart disease [153]. ApoM levels were reported as a potential biomarker for coronary artery disease [154,155]; however, another study did not identify apoM as a predictor of coronary heart events [156]. Reduced circulating apoM is independently associated with adverse outcomes across the spectrum of human heart failure [157]. In human apoM transgenic mice, the ability of HDL to mediate cholesterol efflux from peritoneal mouse macrophages and to protect against LDL oxidation was improved [158]. Hepatocyte-specific apoM transgenic mice had larger plasma HDLs enriched with apoM, cholesteryl ester, LCAT and S1P, however in vivo macrophage reverse cholesterol transport capacity was similar to that of wild-type mice [159]. ApoM-enriched HDL derived from apoM-transgenic mice showed an increased in vitro cholesterol efflux capacity from macrophages compared to HDL derived from wild-type mice [160]. However, apoM had no major effect on the excretion of cholesterol into feces [160]. Moreover, in apoM deficient mice, cholesterol accumulated in large HDL particles and HDL to preβ-HDL conversion was impaired. Cholesterol efflux capacity of apoM-deficient HDL was reduced in vitro, indicating that apoM is important for preβ-HDL formation and cholesterol efflux capacity of HDL [161].

#### 3.3.2. Effects of HDL-Associated Lipids on Macrophage Function

Following endotoxin expression, macrophages express high levels of S1P, which activate in turn S1P receptor 2 and S1P receptor 3, triggering the expression of pro-inflammatory mediators, such as C–C motif chemokine ligand (CCL) 2, IL-1β and IL-18 [162].

In a recent study, a link between anti-apoptotic effects of HDL on macrophages and HDL-S1P content was demonstrated [163]. Specifically, like S1P, HDL induced STAT3 phosphorylation, survivin expression and inhibition of caspase-3 activation. These effects were mimicked by lipids isolated from HDL and by apoM-containing HDL, but not by apoA-I or HDL deprived of S1P and apoM. Pharmacological antagonists of S1P receptors attenuated the anti-apoptotic signaling produced by HDL in macrophages [163]. Another study showed that HDL-S1P and albumin-S1P reduced macrophage adhesion to endothelial cells in vitro [164].

Activation of S1P receptor 1 is involved in macrophage polarization towards an anti-inflammatory phenotype [165]. At the same time, in a thioglycollate peritonitis model, S1P inhibited macrophage migration through S1P receptor 2 ligation [166]. A more recent study demonstrated that HDL stimulated migration of macrophages was dependent on SR-BI and was blocked by S1P receptor antagonists [167]. S1P was recently recognized as an intermediate in liver X receptor (LXR)-stimulated ABCA1-mediated cholesterol efflux. S1P/S1P receptor 3 signaling was identified as a positive feedback regulator of macrophage cholesterol efflux by sphingolipids [168]. In atherosclerosis animal models, S1P receptor 2 deficiency was associated with reduced inflammation and monocyte/macrophage recruitment [162,169]. In contrast, S1P receptor 3 was shown to mediate the chemotactic effect of S1P [170]. S1P binding to the S1P receptor was shown to provoke an anti-inflammatory macrophage phenotype via inhibition of pro-inflammatory cytokine production and NF-κB activation, inhibiting macrophage cell death and increasing cyclic adenosine monophosphate production [66].

#### 3.3.3. Effects of HDL-Associated Enzymes on Macrophage Function

Among the paraoxonase family of enzymes, PON1 and PON3 are mainly associated with HDL [44]. PON1 was shown to directly impact inflammation via attenuation of inflammatory cytokine release of macrophages, such as TNF-α and IL-6 [171]. PON1 treated mice depicted smaller mouse peritoneal macrophages with a lower granulation level than those isolated from control mice [94].

Both PON1 and PON3 are thought to reduce the lipoprotein atherogenicity through hydrolysis of oxidized lipids [172,173], resulting in reduced uptake of atherogenic lipoproteins by macrophages [174]. When macrophages were exposed to HDL in the presence of a PON1 antibody, cholesterol efflux capacity and the ability of HDL to inhibit macrophage-dependent oxidation of LDL were impaired [175]. Similarly, PON1-deficiency in mice resulted in increased oxidative stress both in serum and peritoneal macrophages [176]. Incubation of mouse peritoneal macrophages with HDL derived from PON1 transgenic mice enhanced the cholesterol efflux capacity compared to HDL derived from PON1^−/−^ mice [47]. It has been shown that PON1 interacts with lipid rafts on the plasma membrane [177]. In addition, PON1 was shown to inhibit mouse peritoneal macrophage cholesterol biosynthesis and atherogenesis, potentially through its phospholipase A2-like activity [178]. In addition, the PON1-192R/Q human polymorphism resulted in reduced PON1 stability, lipolactonase activity and macrophage cholesterol efflux, implying a potential role of the polymorphism to atherosclerosis susceptibility [179]. Expression of PON3 in apoE-deficient mice resulted in significantly lower serum levels of lipid hydroperoxides and enhanced macrophage cholesterol efflux potential [180]. Moreover, human paraoxonase gene cluster transgenic overexpression repressed atherogenesis and promoted atherosclerotic plaque stability in apoE-deficient mice [181].

Along with its ability to reduce lipid peroxides in HDL, PON1 was shown to reduce oxidant formation in macrophages [182]. Specifically, PON1 overexpression in an experimental diabetes mouse model was associated with decreased macrophage-associated oxidative stress, decreased diabetes development and mortality [183]. In addition, overexpression of human PON1 in mice with combined leptin and LDL receptor deficiency resulted in a significant reduction of total plaque volume and the volume of plaque macrophages and of plaque-associated oxidized LDL [184].

#### 3.3.4. Effects of Reconstituted HDL and Mimetic Peptides on Macrophage Function

Reconstituted HDL consisting of apoA-I complexed with phosphatidylcholine inhibited TLR2 receptor expression and decreased NF-κB activation and pro-inflammatory cytokine production in human monocyte-derived macrophages [112]. Moreover, infusion of reconstituted HDL in healthy individuals protected from inflammatory events caused by LPS [120], while in type 2 diabetes mellitus patients it increased the capacity of plasma to receive cholesterol from THP-1 macrophages [104]. Discoidal reconstituted HDL containing phosphatidylcholine complexed with apoA-I inhibited reactive oxygen species production, NADPH oxidase activity, p47phox translocation from the cytoplasm to the plasma membrane and NADPH oxidase 2 expression in human macrophages incubated under high glucose [121]. Reconstituted HDL-containing S1P could induce macrophage cholesterol efflux independently of S1P but had additional S1P-mediated effects on endothelial cell tube formation mediated by Akt/ERK/NO through the S1P receptor 2 and S1P receptor 3 [185]. Reconstituted HDL carrying apoE exhibited properties similar to those of HDL carrying apoA-I, but with a lower capacity to stabilize PON1 and to induce its antiatherogenic functions, including inhibition of LDL oxidation and stimulation of macrophage cholesterol efflux [186].

It has been shown that the apoA-I mimetic peptide 4F promoted the M2 macrophage polarization [107]. In addition, the apoA-I mimetic peptide 4F removed cholesterol from lipid rafts. It downregulated TLR cell surface expression in LPS-treated monocyte-derived macrophages, resulting in downregulation of genes modulated by the TLR pathway [107,187]. Oral administration of the apoA-I mimetic peptide 4F in mice promoted forming of preβ-HDL with increased paraoxonase activity, resulting in improved HDL anti-inflammatory properties and cholesterol efflux capacity both in vitro and in vivo [188,189]. Intranasal administration of full-length human apoA-I to house dust mite-challenged mice lead to a decreased number of bronchoalveolar lavage fluid macrophages, associated with a reduction in airway inflammation [31].

The effects of apoB-depleted serum, isolated HDL, reconstituted HDL, HDL-associated apolipoproteins, lipids and enzymes, as well as mimetic peptides on macrophage function, both in human studies and in studies utilizing animal models are summarized in Table 2.

### 3.4. HDL and Neutrophil Function

Neutrophils, the most abundant innate immune cells, are related to chronic inflammation and autoimmune diseases, such as rheumatoid arthritis [190] or psoriasis [191]. Neutrophils are also associated with obesity [192], atherosclerosis [193] and acute coronary events [194,195], with their presence being identified in atherosclerotic lesions [196,197]. Neutrophils can become activated in hyperlipidemia. The severity of the disease is directly correlated with superoxide release and CD11b expression [198,199,200]. Neutrophil activation can be directly triggered by cholesterol loading [201]. The main offensive functions of these cells include the respiratory burst, which is linked to the generation of reactive oxygen species, degranulation and the formation of neutrophil extracellular traps (NETs) [202,203]. NETs are a key component of pathological thrombi and drive cardiovascular, inflammatory and thrombotic diseases in humans and mice [204] and were shown to promote atherosclerosis and carotid thrombosis in *ApoE^−/−^* mice [205,206,207,208]. Moreover, myeloid deficiency of ABCA1 and ABCG1 leads to macrophage and neutrophil inflammasome activation, which in turn promotes atherosclerotic plaque development and NETs forming in plaques [209].

#### 3.4.1. Effects of HDL and HDL-Associated Apolipoproteins on Neutrophil Function

ApoA-I rapidly inhibits neutrophil activation and CD11b expression through ABCA1, while mature HDL suppresses effector responses apparently independent of receptors [210]. ApoA-I was also shown to diminish neutrophil degranulation and superoxide production in response to surface-bound immunoglobulin G and N-formyl-L-methionyl-L-leucyl-phenylalanine (fMLP) [211]. Moreover, apoA-I suppressed neutrophil activation associated with reductions in cellular adhesion, degranulation and oxidative burst [211,212]. At the same time, apoA-I was also able to decrease IL-1β release in LPS stimulated neutrophils [213]. Both apoA-I and HDL attenuated neutrophil adhesion and spreading to activated platelet monolayers [210,212]. Interestingly, also apoA-IV potently decreased neutrophil chemotaxis upon IL-8 stimulation [29].

HDL was shown to stimulate the biogenesis of microRNA-223-3p in neutrophils [214]. MicroRNA-223-3p regulates neutrophil development, hyperactivity and recruitment during infection [214]. Another recent study proposed that dysfunctional HDL may contribute to the systemic inflammation in uremic patients via modulation of polymorphonuclear cells’ functions, such as attenuation of apoptosis [215]. Moreover, both apoA-I and HDL decreased neutrophil membrane lipid rafts, which is likely a key event since lipid raft abundance has been correlated with CD11b activation [76]. In fact, many studies have described the importance of lipid rafts not only in neutrophil activation but also in the release of inflammatory mediators [216,217,218,219]. Cholesterol loading of neutrophils is priming their activation and is increasing their endothelium adhesiveness [201].

ApoA-II decreased producing of IL-8 released by neutrophils stimulated either with the acute phase protein SAA or with LPS [213]. The addition of recombinant SAA caused an increase in the basal liberation of TNF-α, IL-1β and IL-8 by human blood neutrophils. In contrast, HDL-associated SAA did not show these activities [220].

#### 3.4.2. Effects of HDL-Associated Lipids on Neutrophil Function

Modification of HDL by secretory phospholipase A2 (sPLA2) results insaturated lysophosphatidylcholines forming. Interestingly, sPLA2 modified HDL (HDL enriched with lysophosphatidylcholines) depicted a dramatically increased ability to suppress agonist-induced neutrophil activation, including shape change, CD11b activation, NET formation, adhesion under flow and migration of neutrophils, when compared to control HDL [71]. This NETosis-preventing effect may be due to the potent lipid raft disrupting capacity of sPLA2-modified HDL and the suppression of intracellular Ca^2+^ rise [71]. Moreover, the HDL-associated lysophosphatidylcholine 16:0 and lysophosphatidylserine 18:0 could inhibit neutrophil shape change, whereas unsaturated lysophosphatidylcholine 18:1 showed no effect [71].

In addition, in a mouse model of myocardial ischemia/reperfusion injury, HDL-associated sphingosylphosphorylcholine reduced infarct size and polymorphonuclear neutrophil recruitment to the infarcted area via the S1P receptor 3 [221]. Similarly, HDL-associated S1P reduced infarct size in a mouse model of myocardial ischemia/reperfusion by inhibiting cardiomyocyte apoptosis and neutrophil recruitment to the infarct area dependent on nitric oxide and the S1P receptor 3 [164]. Moreover, smaller myocardial infarcts and reduced neutrophil infiltration into the infarcted area were observed in apoM (major plasma carrier of S1P) transgenic mice [222].

#### 3.4.3. Effects of Reconstituted HDL and Mimetic Peptides on Neutrophil Function

Reconstituted HDL containing apoA-I and phosphatidylcholine potently decreased cell adhesion via blockage of LPS activity and modification of CD11b/CD18 upregulation [223]. Interestingly, phosphatidylcholine alone was shown to be sufficient for lipopolysaccharide-binding protein catalyzed neutralization of LPS [224]. Infusion of reconstituted HDL in type 2 diabetes mellitus patients reduced neutrophil adhesion to the fibrinogen matrix [104]. In peripheral vascular disease patients, infusion of reconstituted HDL attenuated neutrophil activation [210]. Administration of apoA-I or reconstituted HDL containing apoA-I (or the 5A apoA-I mimetic peptide) complexed with phosphatidylcholine showed potent antiatherogenic effects and reduced the collar-mediated increase in endothelial expression of the cell adhesion molecules VCAM-1 and ICAM-1 in New Zealand white rabbits. In addition, it suppressed the production and expression of the catalytic NADPH oxidase-4 subunits of NADPH oxidase and markedly impaired the infiltration of circulating neutrophils into the carotid intima-media [225,226,227]. ApoA-I promoted atherosclerosis regression in diabetic mice by suppressing myelopoiesis and plaque inflammation [105].

Administration of the apoA-I mimetic peptide 5A in an experimental murine model of house dust mite-induced asthma resulted in a significant reduction of airway inflammation, hyperreactivity and remodeling, as well as in a reduction of bronchoalveolar lavage fluid neutrophils [228]. Administration of the 5A peptide to ovalbumin-challenged apoA-I knockout mice suppressed increases in neutrophilic airway inflammation [229], while administration of L-4F to wild-type mice receiving inhaled LPS reduced the number of bronchoalveolar lavage fluid neutrophils [230]. Moreover, L-4F inhibited the activation of isolated human leukocytes and neutrophils by acute respiratory distress syndrome serum and LPS in vitro [231]. In addition, infusion of recombinant apoA-I-Milano in a transient middle cerebral artery occlusion stroke rat model significantly reduced infarct volume through inhibition of platelet aggregation. Still, it did not reduce hemorrhagic transformation and activation of neutrophils [232].

Intranasal administration of full-length human apoA-I to house dust mite-challenged mice lead to a reduction in airway inflammation, with decreased number of bronchoalveolar lavage fluid neutrophils [31]. ApoA-I suppressed the expression of ICAM-1 on endothelium, thus diminishing neutrophil adherence and transendothelial migration and the subsequent myocyte injury in an experimental rat model of ischemia/reperfusion injury [233]. In an experimental mouse model of LPS-induced inflammation and lethality, apoA-I gene transfer resulted in a significantly attenuated LPS-induced infiltration of neutrophils into the lungs, as well as in reduced lung edema and mortality [234]. A single low dose infusion of apoA-I administered after the onset of acute inflammation in carotid arteries of normocholesterolemic New Zealand White rabbits decreased neutrophil infiltration and inhibited their activation [227]. Infusion of lipid-free apoA-I or discoidal reconstituted HDL containing phosphatidylcholine and apoA-I decreased neutrophil infiltration and VCAM-1 and ICAM-1 expression in a model of acute vascular inflammation in New Zealand White rabbits [235].

The effects of HDL, reconstituted HDL, HDL-associated apolipoproteins, lipids and enzymes, as well as mimetic peptides on neutrophil function, in human studies and studies utilizing animal models are summarized in Table 3.

### 3.5. HDL and Eosinophil Function

Eosinophil-rich inflammation has long been associated with allergic inflammation, asthma and parasitic infestation. Eosinophils release basic proteins that are cytotoxic and lipid mediators, such as cysteinyl leukotrienes, which cause bronchial epithelial damage and airflow obstruction [236]. Evidence from animal models of asthma and clinical studies demonstrated a causal role of eosinophils in the pathogenesis of asthma, including airway hypersensitivity, remodeling and elevated mucus production [237]. The number of eosinophils increases in several diseases, including helminth infections, hypereosinophilic syndrome, allergies [237] and acute myocardial infarction [238], while eosinophil levels have emerged as a strong predictor of mortality in acute heart failure [239] and coronary artery disease patients [240]. Granules of mature eosinophils contain basic proteins, such as eosinophil cationic protein, eosinophil peroxidase and eosinophil-derived neurotoxin [241]. In contrast, deposition of granules released from eosinophils in tissues comprises a common finding in eosinophil-associated diseases and potentially contributes to their pathogenesis [242,243,244,245]. Recently, however, it was recognized that eosinophils are crucial for local immunity and repair, with an increasing number of regulatory and homeostatic roles attributed to them. An important function of eosinophils is their antitumor effect in colorectal cancer [246]. Eosinophils show hepatoprotective activity [247] and cardiac protective function after myocardial infarction [248]. Of particular interest, a robust inverse correlation between eosinophil numbers and coronavirus disease 2019 (COVID-19) infection severity was observed most recently [249]. Taken together, these new findings point to an unmet need to target eosinophil overactivation without completely depleting this multifunctional immune cell type. Therefore, it is important to investigate whether HDL or an HDL-associated component could serve as a potential new target to reduce eosinophil activation.

#### 3.5.1. Effects of HDL and HDL-Associated Components on Eosinophil Function

In coronary artery disease patients, an inverse association of absolute eosinophil count and HDL cholesterol and a positive association with the prevalence of coronary artery disease was reported [250]. Both HDL and HDL apolipoproteins were recently shown to effectively inhibit eosinophil chemotaxis [29] and to attenuate eosinophil activation [251]. In a study involving atopic dermatitis patients, patients’-derived HDL showed an impaired ability to inhibit agonist-induced eosinophil shape change and migration compared to HDL isolated from healthy controls [34]. In contrast, an increased ability of isolated HDL derived from allergic rhinitis patients to suppress eosinophil effector responses upon eotaxin-2/CCL24 stimulation was demonstrated [28]. Importantly, apoA-IV applied at very low concentrations, decreased eosinophil shape change, chemotaxis, CD11b expression and Ca^2+^ flux. The molecular mechanism involved the activation of Rev-ErbA-α followed by the induction of a phosphatidylinositol-3-kinase (PI3K)/phosphoinositide-dependent-kinase 1 (PDK1)/protein kinase A (PKA)-dependent signaling cascade [29]. In addition, apoA-IV could accelerate eosinophil apoptosis of allergic donors, while apoA-I was less effective [29]. Interestingly, besides apoA-IV and apoA-I, apoC-III effectively and dose-dependently suppressed agonist-induced eosinophil shape change [34]. Moreover, intranasal administration of full-length human apoA-I to house dust mite-challenged mice lead to a reduction in airway inflammation, with decreased number of bronchoalveolar lavage fluid eosinophils [31]. Another study evaluated the role of HDL in lung-allergic inflammation of ovalbumin-challenged endothelial lipase knockout mice. A reduction in the number of eosinophils in bronchoalveolar lavage and in the expression of VCAM-1, as well as an attenuation of hyperresponsiveness, was shown in endothelial lipase knockout mice. This indicated that targeted inactivation of endothelial lipase attenuated lung-allergic inflammation. At the same time, the protective effects were associated with high plasma HDL levels, downregulation of VCAM-1 and loss of the direct ligand-binding function of endothelial lipase [252].

In addition to HDL apolipoproteins, the major HDL-associated saturated lysophosphatidylcholine species 16:0 and 18:0 were shown to effectively and dose-dependently inhibit agonist-induced shape change and migration of eosinophils [34]. Along with this, another study demonstrated that lysophosphatidylcholines suppressed multiple eosinophil effector responses, such as CD11b upregulation, chemotaxis, degranulation and downstream signaling and suppressed eosinophil migration in vivo [72]. In an experimental murine model of house dust mite extract-induced asthma, apoA-IV could repress the infiltration of eosinophils into the bronchoalveolar space and protected mice from the airway and systemic eosinophilia [29], while lysophosphatidylcholine 18:0 treatment markedly reduced immune cell infiltration into the lungs in a mouse model of allergic cell recruitment [72]. Interestingly, the stable lysophosphatidylcholine analog miltefosine also showed very similar properties, suppressing human eosinophil activation and ameliorating murine allergic inflammation in vivo [253].

#### 3.5.2. Effects of Mimetic Peptides on Eosinophil Function

It has been reported that the apoA-I/ABCA1 pathway may have a protective effect on asthma, supporting the concept of advancing inhaled apoA-I mimetic peptides to a clinical trial of asthma [254]. Specifically, administration of the 5A apoA-I mimetic peptide in an experimental murine model of house dust mite-induced asthma resulted in a significant reduction of bronchoalveolar lavage fluid eosinophils [228]. Similarly, intranasal administration of D-4F, another apoA-I mimetic peptide, reduced airway eosinophilia and airway resistance in ovalbumin-challenged mice [255].

The effects of apoB-depleted serum, isolated HDL, HDL-associated apolipoproteins and lipids, as well as mimetic peptides on eosinophil function, both in human studies and in studies utilizing animal models are summarized in Table 4.

### 3.6. HDL and Dendritic Cell Function

Dendritic cells comprise a heterogenous family of bone marrow-derived immune cells of both lymphoid and myeloid stem cell origin that populate all lymphoid organs, including the spleen, thymus and lymph nodes, as well as almost all nonlymphoid organs and tissues [256]. They are responsible for the process and presentation of antigens to naïve, memory and effector T cells [73,256]. At the same time, they are implicated in the pathogenesis of autoimmune diseases, such as psoriasis [257,258] and systemic lupus erythematosus [256], as well as allergies, including allergic rhinitis [259], allergic asthma [260] and atopic dermatitis [261]. It was suggested that dendritic cells are critically involved in the progression and destabilization of atherosclerotic plaques [262,263]. In contrast, in atherosclerotic plaques, it has been shown that plasmacytoid dendritic cells stimulate T cells against viral antigens [264]. The exact mechanisms of action of dendritic cells, along with their role in immunity and their implication in diseases, have been described in detail elsewhere and are not in the focus of the current review [256,265].

#### Effects of HDL and HDL-Associated Components on Dendritic Cell Function

It has been shown that hyperlipidemia altered dendritic cell function, specifically by inhibiting cell migration. At the same time, HDL and HDL-associated PAF-AH restored this process [266]. HDL, along with some of its components, were shown to interfere with certain steps of dendritic cells’ activity and maturation. Specifically, apoA-I impaired adaptive immunity via inhibition of maturation, differentiation and function of dendritic cells [267,268,269] by inducing prostaglandin E2 and IL-10, two known inhibitors of dendritic cell function and differentiation, and by inhibiting the ability of dendritic cells to secrete IL-12 when stimulated with anti-CD40 and IFN-γ [268]. HDL was also able to reduce IL-12 production in stimulated mature dendritic cells, thus decreasing their ability to stimulate T cells [268,270]. Moreover, upon LPS-mediated TLR4 stimulation, HDL inhibited the ability of dendritic cells to induce Th1 response. At the same time, the phospholipid HDL fraction was identified as the most active in inhibiting dendritic cell maturation [270]. Specifically, HDL-associated 1-palmitoyl-2-linoleoyl-phosphatidylcholine and 1-stearoyl-2-linoleoyl-phosphatidylcholine were shown to have direct immunoregulatory functions by impairing the ability of dendritic cells to activate a Th1 response of T cells [270]. Moreover, reconstituted HDL particles could trigger immunogenic cell death and promoted dendritic cell maturation in an experimental model of hepatocellular carcinoma [271]. At the same time, it has been shown that oxidized HDL may promote the maturation and migration of bone marrow-derived dendritic cells in vitro [272].

The effects of HDL, HDL-associated enzymes, as well as reconstituted or synthetic HDL on dendritic cell function, both in human studies and in studies utilizing animal models are summarized in Table 5.

S1P is a major regulator of both dendritic cell activation and maturation [273]. S1P effectively diminished the ability of dendritic cells to capture antigens via macropinocytosis, while most studies supported that extracellular S1P presence on dendritic cells leads to an IL-6, IL-23, STAT3-dependent T helper type 17 inflammatory profile; although at least in part Th1 was attenuated [273]. Topical application of S1P was shown to be beneficial in atopic dermatitis treatment [274]. At the same time, S1P homeostasis dysregulation has been discussed in the pathogenesis of the disease. However, in systemic inflammatory syndromes, such as bacterial sepsis and viral hemorrhagic fever, S1P promoted the dissemination of inflammation by contributing to the coagulation-induced activation and trafficking of dendritic cells in the lymphatics [275].

### 3.7. Role of HDL and HDL-Associated Components on T Cell Function

The allergic response is engineered by CD4+ T lymphocytes secreting Th2 cytokines upon activation by allergen-derived peptides [276]. At the same time, immune-mediated skin diseases may be mediated mainly by T cells through uncontrolled, unspecific inflammation and via the humoral immune system [277]. Importantly, T cell activation plays a critical role in the pathogenesis of psoriasis [278], which is T17/T22 cell-dominated, and atopic dermatitis, a T2 cell-dominated disease [277].

HDL-induced cholesterol efflux from macrophages affected antigen presentation to T cells, along with T cell receptor signaling [267,279,280]. At the same time, the HDL concentration regulated cellular contact between stimulated T cells and monocytes [281]. HDL-associated apoA-I inhibited producing of IL-1β and TNF-α by blocking the contact-mediated activation of monocytes by T lymphocytes through its binding to stimulated T cells [282], while HDL potently reduced reactive oxygen species production induced in polymorphonuclear neutrophils upon contact with stimulated T cells [283].

ApoA-I was shown to control the cholesterol-associated T-lymphocyte activation and proliferation in peripheral lymph nodes of diet-fed LDLr^−/−^, apoA-I^−/−^ mice [284] and to suppress inflammation through stimulation of regulatory T cells (Tregs) in the lymph nodes and through inhibition of effectors, such as memory T cells [285]. Tregs could specifically internalize HDLs from their microenvironment and use them as an energy source, a fact likely attributable to the increased SR-BI cell expression. At the same time, HDLs could significantly decrease the apoptosis of human Tregs in vitro [286].

ApoA-II was shown to suppress IFN-γ production by concanavalin A-stimulated human CD4 T cells and to attenuate concanavalin A-induced hepatitis. Therefore, apoA-II could be an effective therapeutic agent for CD4 T cell-dependent autoimmune or viral human hepatitis [287].

In a study evaluating PON1 activity in individuals infected with human immunodeficiency virus (HIV) type-1, it has been shown that the enzyme activity was correlated with the number of CD4+ T cells, suggesting an association of PON1 with the immune status of HIV type-1 infected individuals [288]. Along with this, another group demonstrated impaired PON1 activity in HIV patients compared to controls. At the same time, HIV infection was associated with functional and compositional HDL alterations associated with CD4+ T cell counts [289].

Importantly, the S1P gradient and the cell surface residence of S1P receptor 1 on T cells are two key factors that mediate lymphocyte egress from peripheral lymphoid organs and the thymus [290,291,292]. In addition, S1P was reported to reduce T cell apoptosis [293]. The S1P receptor 1 expression was associated with T cell activation status [294,295] and lineage determination [296]. Specifically, S1P inhibited forkhead box P3 (FoxP3)+ Tregs differentiation, while it reciprocally promoted Th1 development [296]. S1P receptor antagonized transforming growth factor-beta receptor function through inhibition of small mother against decapentaplegic homolog 3 (SMAD3) activity to control Tregs and Th1 dichotomy [296]. Finally, FTY720, a synthetic S1P analog, was shown to inhibit atherosclerosis via modulation of lymphocyte and macrophage function, which is consistent with the notion that S1P contributes to the antiatherogenic potential of HDL [297,298].

## 4. Conclusions

HDL composition, function and plasma levels have been associated with altered immune responses. Accumulating evidence suggests an important modulatory ability of HDL particles, purified HDL-associated proteins, and lipids in the activation state and function of immune cells. It has long been known that HDL plays an anti-inflammatory role in inflammation and infection. At the same time, more recent studies also provided evidence for the role of HDLs in allergy and atopic skin diseases. In addition, alterations in the ability of HDL to modulate immune cell apoptosis, activation, chemotaxis, expression of cell surface markers and pro-inflammatory cytokine secretion were observed. Such alterations could have a major impact on disease progression and affect the risk for infections and cardiovascular disease. Several groups over the years have attempted to demonstrate, both in in vitro and in vivo experiments, which HDL components are primarily responsible for the anti-inflammatory and anti-allergic effects. Of particular interest, purified apoA-I, apoA-IV and lysophosphatidylcholine could suppress neutrophil activation, adhesion and chemotaxis. At the same time, apoA-I, apoA-IV, apoC-III and lysophosphatidylcholine effectively inhibited eosinophil activation and function. ApoA-I was also shown to promote macrophage M2 polarization and cholesterol efflux. Moreover, it suppressed reactive oxygen species production, TLR expression and activation of inflammatory response in macrophages, along with dendritic cell maturation, differentiation and function. Although apoC-III effectively suppressed eosinophil shape change, it induced adhesion and inflammasome activation on monocytes. At the same time, it increased vascular adhesion molecules expression of endothelial cells. Moreover, HDL-associated paraoxonase was shown to affect monocyte and macrophage expression of cell surface markers, adhesion, chemotaxis and inflammatory cytokine release. A summary of the effects of HDL-associated or purified apolipoproteins, lipids and enzymes in immune cell activation and function in vitro is given in Figure 1.

Along with this, apoA-I mimetic peptides, including the 5A-peptide and the 4F-peptide, were shown to decrease activation, infiltration, neutrophilic airway inflammation and airway eosinophilia as well as to attenuate monocyte/macrophage TLR cell surface expression and signaling pathway. A summary of the different apolipoprotein mimetic peptides known to affect immune cell function and, therefore, mentioned in this review is given in Table 6.

Importantly, apart from the aforementioned mimetic peptides, administration of apoA-I_Milano_ nanoparticles has gained much attention for treating heart failure and coronary artery disease [303,304,305]. Briefly, apoA-I_Milano_ is an apoA-I mutant resulting from an arginine 173 to cysteine mutation [306,307], leading to a higher life expectancy in heterozygotes and a lower atherosclerosis rate [304]. MDCO-216 is a form of reconstituted HDLs consisting of purified recombinant dimer apoA-I_Milano_ complexed with 1-palmitoyl-2-oleoyl-sn-glycero-3-phosphatidylcholine [308]. In mice with pre-existing heart failure, treatment with MDCO-216 induced regression of interstitial fibrosis, normalization of lung weight, improved isovolumetric relaxation and increased relative myocardial vascularity [303]. The efficacy of MDCO-216 was also demonstrated in a mouse model of hypertension-associated heart failure with preserved ejection fraction [305].

Moreover, reconstituted forms of HDL have already been applied in clinical use to attenuate atherosclerotic vascular disease and to reduce cardiovascular risk [309]. At the same time, their potent anti-inflammatory properties can also be exploited to reduce inflammation in diseases such as rheumatoid arthritis and type 2 diabetes [310]. Specifically, reconstituted HDL particles, mainly apoA-I and phosphatidylcholine, could effectively decrease neutrophil activation and adhesion in type 2 diabetes and peripheral vascular disease patients. At the same time, they also effectively decreased monocyte CD11b expression in type 2 diabetes patients. In addition, they effectively inhibited macrophage reactive oxygen species production, pro-inflammatory cytokine secretion and TLR expression. At the same time, they promoted cholesterol efflux from macrophages.

On the other hand, most diseases strongly influence the metabolism, composition and subsequent functionality, such as immunomodulatory functions of HDL. This leads in most cases to impaired HDL functionality, such as cholesterol efflux capacity, the ability of HDL to modulate immune cell activation, chemotaxis, expression of cell surface markers and pro-inflammatory cytokine secretion. Such alterations could have a major impact on disease progression and affect the risk for infections and cardiovascular disease.

To conclude, HDL and its associated components appear to have a major impact on the modulation of immune cell activation status and various aspects of immune cell function and comprise a promising tool for future therapeutic interventions.

## Figures and Tables

**Figure 1 biomedicines-09-00587-f001:**
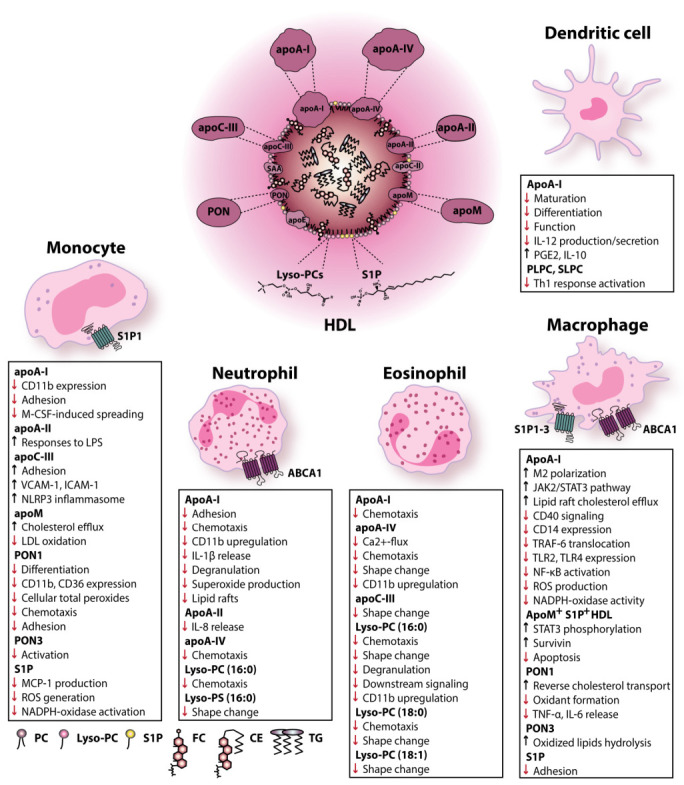
Effects of HDL-associated or purified apolipoproteins, lipids and enzymes in immune cell activation and function in vitro. Abbreviations: ABCA1—ATP-binding cassette subfamily A member 1; apoA-I—apolipoprotein A-I; apoA-II—apolipoprotein A-II; apoA-IV—apolipoprotein A-IV; apoC-II—apolipoprotein C-II; apoC-III—apolipoprotein C-III; apoE—apolipoprotein E; apoM—apolipoprotein M; CD—cluster of differentiation; CE—cholesteryl ester; FC—free cholesterol; HDL—high-density lipoprotein; ICAM-1—intercellular adhesion molecule 1; IL—interleukin; JAK2—Janus kinase 2; LDL—low-density lipoprotein; LPS—lipopolysaccharide; MCP-1—monocyte chemoattractant protein-1; M-CSF—macrophage colony-stimulating factor; NADPH—nicotinamide adenine dinucleotide phosphate; NLRP3—nod-like receptor family pyrin domain-containing 3; PC—phosphatidylcholine; PGE2—prostaglandin E2; PLPC—1-palmitoyl-2-linoleoyl-phosphatidylcholine; PON—paraoxonase; ROS—reactive oxygen species; S1P—sphingosine-1-phosphate; S1P1—sphingosine-1-phosphate receptor 1; S1P2—sphingosine-1-phosphate receptor 2; S1P3—sphingosine-1-phosphate receptor 3; SAA—serum amyloid A; SLPC—1-stearoyl-2-linoleoyl-phosphatidylcholine; STAT3—signal transducer and activator of transcription 3; TG—triglyceride; Th1—T helper type 1; TLR—Toll-like receptor; TNF-α—tumor necrosis factor α; TRAF-6—TNF receptor-associated factor 6; VCAM-1—vascular cell adhesion molecule 1.

**Table 1 biomedicines-09-00587-t001:** Effects of apoB-depleted serum, HDL, reconstituted HDL, HDL-associated apolipoproteins or mimetic peptides on monocyte function in human studies utilizing primary monocytes or a monocyte cell line and in studies utilizing animal models.

ApoB-Depleted Serum, HDL, HDL-Associated Protein, rHDL, Mimetic Peptide	Human Study/Animal Model/Cell Line	Effect on Monocytes	References
Human studies			
apoB-depleted serum, HDL	Allergic rhinitis patients/psoriasis patients under biologic treatment, U937 cell line	Decreased anti-inflammatory potential	[28,89]
HDL	Human monocytes/endothelial cells	Decreased CD11b activation, adhesion, chemotaxis, spreading	[74,75,76]
rHDL-containing apoA-I and PC	Type 2 diabetes patients monocytes	Decreased CD11b expression	[104]
4F-peptide	Human monocytes, THP-1 cell line	Promoted M2 polarization, attenuated TLR4, CD14 and lipid raft expression	[107]
Ac-hE18A-NH_2_-peptide	Human umbilical vein endothelial cells, monocytes	Decreased adhesion, IL-6, MCP-1 secretion, VCAM-1 expression	[106]
Animal studies			
HDL/apoA-I	Diabetic apoA-I-Tg mouse model	Improved cholesterol efflux, suppressed proliferation and monocyte production	[105]

A summary of the effects of the apoB-depleted serum, HDL, reconstituted HDL, HDL-associated apolipoproteins, as well as apoA-I and apoE mimetic peptides on monocyte activation and functional properties is given, as described from human studies or studies utilizing animal models. Abbreviations: apoA-I—apolipoprotein A-I; apoB—apolipoprotein B—CD—cluster of differentiation; HDL—high-density lipoprotein; IL-6—interleukin 6; MCP-1—monocyte chemoattractant protein-1; PC—phosphatidylcholine; rHDL—reconstituted high-density lipoprotein; Tg—transgenic; TLR4—Toll-like receptor 4; VCAM-1—vascular cell adhesion molecule 1.

**Table 2 biomedicines-09-00587-t002:** Effects of apoB-depleted serum, HDL, reconstituted HDL, HDL-associated apolipoproteins, lipids and enzymes or mimetic peptides on macrophage function in human studies utilizing monocyte-derived macrophages or cell lines and in studies utilizing animal models.

ApoB-Depleted Serum, HDL, HDL-Associated Protein/ Lipid/Enzyme, rHDL, Mimetic Peptide	Human Study/Animal Model/ Cell Line	Effect on Macrophages	References
Human studies			
apoB depleted serum, HDL	Atrial fibrillation, psoriasis, acute coronary syndrome or renal disease patients, J774.2 and RAW 264.7 macrophages	Impaired cholesterol efflux capacity	[13,14,23,89,150,151]
LpA-I/A-II HDL particles	RAW 264.7 macrophages	ApoA-II presence in HDL particles enhanced ABCA1-mediated cholesterol efflux compared to LpA-I particles	[124]
HDL, rHDL or commercially obtained HDL	Human bone-marrow-derived macrophages	Increased gene and protein expression of pro-inflammatory IL-12 and TNF-α and decreased anti-inflammatory IL-10 via a mechanism involving lipid raft disruption and PKC	[114]
HDL, rHDL-containing apoA-I and PC	Human monocyte-derived macrophages	Inhibited ROS production, NADPH oxidase activity, Nox2 expression	[121]
rHDL-containing apoA-I and PC	Human monocyte-derived macrophages	Inhibited TLR2 expression, decreased NF-κB activation and pro-inflammatory cytokine production	[112]
rHDL-containing apoA-I and PC	Type 2 diabetes patients, THP-1 macrophage-derived foam cells	Improved plasma cholesterol efflux capacity	[104]
4F-peptide	Human monocyte-derived macrophages	Promoted M2 polarization, attenuated TLR4, CD14 and lipid raft expression	[107]
4F-peptide	Human monocyte-derived macrophages, THP-1 macrophage-derived foam cells	Depleted cholesterol from lipid rafts, downregulated TLR cell surface expression and signaling pathway	[107,187]
Animal studies			
HDL	Atherosclerosis regression mouse model	Increased M2 macrophage markers	[109]
HDL	Mouse peritoneal macrophages	Suppressed TRAM/TRIF arm of TLR4 signaling	[113]
HDL	Bone-marrow-derived macrophages/peritoneal macrophages derived from apoA-I Tg mice	Enhanced TLR1/2, TLR3, TLR4, TLR7/8 and TLR9 responses	[114]
HDL	Human apoA-II-Tg Japanese white rabbit model	Stronger cholesterol efflux capacity and inhibitory effects on inflammatory cytokine secretion by macrophages	[145]
HDL	Human apoA-II-Tg mice	Human apoA-II maintained effective reverse cholesterol transport from macrophages to feces despite an HDL deficiency. Increased mice plasma ability to extract cholesterol from macrophages	[146,147]
apoA-I	House dust-mite mouse model	Decreased airway inflammation and number of bronchoalveolar lavage fluid macrophages	[31]
HDL, apoM-enriched HDL	apoM-Tg mouse model	Improved cholesterol efflux capacity and protection against LDL oxidation	[158,160]
apoM-deficient HDL	apoM-deficiency mouse model	Impaired cholesterol efflux capacity	[161]
D-4F-peptide	apoE-deficiency mouse model	Improved HDL-mediated cholesterol efflux	[189]
PON1-Tg mouse model-derived HDL	Mouse peritoneal macrophages	Improved cholesterol efflux capacity	[47]
PON1	PON1-deficiency mouse model-derived macrophages	Decreased cellular peroxide content, superoxide anion release and oxidation of LDL	[176]
PON1	PON1-deficiency mouse model-derived peritoneal macrophages	Inhibited cholesterol biosynthesis and atherogenesis	[178]
PON1	PON1-Tg diabetes mouse model	Decreased diabetes-induced macrophage oxidative stress	[183]
PON3	apoE-deficiency mouse model	Decreased lipid hydroperoxides, improved macrophage cholesterol efflux capacity	[180]
HDL-S1P	Myocardial ischemia/reperfusion mouse model	Decreased adhesion to endothelial cells	[164]
rHDL-containing apoA-I, PC and S1P	RAW264 macrophages	Induced cholesterol efflux	[185]

A summary of the effects of apoB-depleted serum, HDL, reconstituted HDL, HDL-associated apolipoproteins, lipids and enzymes, as well as apoA-I mimetic peptides on macrophage activation and functional properties is given, as described from human studies utilizing monocyte-derived macrophages or studies utilizing animal models. Abbreviations: apoA-I—apolipoprotein A-I; apoB—apolipoprotein B—apoE—apolipoprotein E; apoM—apolipoprotein M; CD—cluster of differentiation; HDL—high-density lipoprotein; LDL—low-density lipoprotein; NADPH—nicotinamide adenine dinucleotide phosphate; NF-κB—nuclear factor-κΒ; Nox2—nicotinamide adenine dinucleotide phosphate oxidase 2; PC—phosphatidylcholine; PKC—protein kinase C; PON1—paraoxonase 1; PON3—paraoxonase 3; rHDL—reconstituted high-density lipoprotein; ROS—reactive oxygen species; S1P—sphingosine-1-phosphate; Tg—transgenic; TLR—Toll-like receptor; TLR1—Toll-like receptor 1; TLR2—Toll-like receptor 2; TLR3—Toll-like receptor 3; TLR4—Toll-like receptor 4; TLR7—Toll-like receptor 7; TLR8—Toll-like receptor 8; TLR9—Toll-like receptor 9; TRAM—TRIF-related adaptor molecule; TRIF—TIR-domain-containing adapter-inducing interferon-β.

**Table 3 biomedicines-09-00587-t003:** Effects of HDL, reconstituted HDL, HDL-associated apolipoproteins, lipids and enzymes or mimetic peptides on neutrophil function in human studies utilizing primary neutrophils and in studies utilizing animal models.

HDL, HDL-Associated Protein/Lipid/Enzyme, rHDL, Mimetic Peptide	Human Study/Animal Model	Effect on Neutrophils	References
Human studies			
HDL	Uremic patients, human neutrophils	Decreased apoptosis	[215]
rHDL-containing apoA-I and PC	Type 2 diabetes patients	Decreased adhesion	[104]
rHDL	Peripheral vascular disease patients	Decreased activation	[210]
rHDL-containing apoA-I and PC	Human polymorphonuclear and endothelial cells	Decreased adhesion via LPS blocking and modification of CD11b/CD18	[223]
L-4F-peptide	Human neutrophils	Decreased activation	[231]
Secretory PLA2-modified HDL	Human neutrophils	Decreased shape change, chemotaxis, adhesion, CD11b activation, NET formation	[71]
Animal studies			
apoA-I, rHDL-containing apoA-I, 5A-peptide complexed with PC	New Zealand white rabbits	Decreased infiltration of circulating neutrophils into carotid intima-media	[225,226,227]
apoA-I, rHDL-containing apoA-I and PC	New Zealand white rabbits	Decreased neutrophil infiltration, VCAM-1 and ICAM-1 expression	[235]
apoA-I/HDL overexpression	Diabetic mice	Decreased neutrophil production and NETs	[105]
5A-peptide	Asthma mouse model	Decreased bronchoalveolar lavage fluid neutrophils	[228]
5A-peptide	OVA-challenged apoA-I^−/−^ mice	Decreased neutrophilic airway inflammation	[229]
L-4F-peptide	LPS-challenged WT mice	Decreased bronchoalveolar lavage fluid neutrophils	[230]
HDL-SPC	S1P3^−/−^ myocardial ischemia/reperfusion mice	Decreased infarct size and neutrophil apoptosis/recruitment	[221]
HDL-S1P	Mouse model of ischemia/reperfusion	Decreased neutrophil recruitment in the infarcted area	[164]

A summary of the effects of HDL, reconstituted HDL or HDL-associated apolipoproteins, lipids and enzymes, along with apoA-I mimetic peptides on neutrophil activation and functional properties is given, as described from human studies, studies utilizing primary neutrophils or studies utilizing animal models. Abbreviations: apoA-I—apolipoprotein A-I; CD—cluster of differentiation; HDL—high-density lipoprotein; ICAM-1—intercellular adhesion molecule 1; LPS—lipopolysaccharide; NET—neutrophil extracellular trap; OVA—ovalbumin; PC—phosphatidylcholine; PLA2—phospholipase A2; rHDL—reconstituted high-density lipoprotein; S1P—sphingosine-1-phosphate; S1P3—sphingosine-1-phosphate receptor 3; SPC—sphingosylphosphorylcholine; VCAM-1—vascular cell adhesion molecule 1.

**Table 4 biomedicines-09-00587-t004:** Effects of apoB-depleted serum, HDL, HDL-associated apolipoproteins and lipids or mimetic peptides on eosinophil function in human studies utilizing primary eosinophils and in studies utilizing animal models.

ApoB-Depleted Serum, HDL, HDL-Associated Protein/Lipid, Mimetic Peptide	Human Study/Animal Model	Effect on Eosinophils	References
Human studies			
apoB-depleted serum, HDL	Allergic rhinitis patients, human eosinophils	Inhibited shape change and chemotaxis	[28]
HDL	Atopic dermatitis patients, human eosinophils	Decreased ability to inhibit shape change and chemotaxis	[34]
HDL, apoA-I, apoA-IV	Allergic patients, human eosinophils	Decreased chemotaxis, accelerated apoptosis	[29]
Stable LPC analog Miltefosine	Human eosinophils	Inhibited shape change, CD11b expression, chemotaxis, degranulation, CD63 expression and Ca^2+^ flux	[253]
Animal studies			
apoA-I, apoA-IV, 5A-peptide	House dust mite-induced asthma mouse model	Decreased bronchoalveolar lavage fluid eosinophils	[29,31,228]
D-4F	OVA-challenged mouse model	Decreased airway eosinophilia	[255]
LPC 18:0	Allergic cell recruitment mouse model	Decreased infiltration into the lungs	[72]
Stable LPC analog Miltefosine	Allergic cell recruitment; allergic lung inflammation mouse models	Suppressed eosinophil migration into the bronchoalveolar lavage; reduced eosinophil numbers, improved lung resistance	[253]

A summary of the effects of apoB-depleted serum, HDL, HDL-associated apolipoproteins and lipids, as well as apoA-I mimetic peptides on eosinophil activation and functional properties is given, as described from human studies utilizing primary eosinophils or studies utilizing animal models. Abbreviations: apoA-I—apolipoprotein A-I; apoA-IV—apolipoprotein A-IV; apoB—apolipoprotein B; CD—cluster of differentiation; HDL—high-density lipoprotein; LPC—lysophosphatidylcholine; OVA—ovalbumin.

**Table 5 biomedicines-09-00587-t005:** Effects of HDL, reconstituted HDL, synthetic HDL or HDL-associated enzymes on dendritic cell function in human studies utilizing monocyte-derived dendritic cells and in studies utilizing animal models.

HDL, HDL-Associated Enzyme, rHDL, sHDL	Human Study/Animal Model	Effect on Dendritic Cells	References
Human studies			
HDL	Human dendritic cells	Impaired ability to activate T cells, decreased IFN-γ, IL-12 and TNF-α secretion	[270]
Animal studies			
HDL, HDL-PAF-AH	ApoE/LDL-deficiency mouse model	Increased migration, restored immunologic priming	[266]
rHDL-containing apoA-I and PC	Mouse BMDCs	Decreased MHC class II, CD40, CD80 and CD86 expression and IL-6, IL-8, IL-12, IL-23, TNF-α and IL-10 secretion; decreased Myd88 mRNA levels	[269]
sHDL	BMDCs from a hepatocellular carcinoma mouse model	Decreased tumor burden triggered immunogenic cell death and induced maturation of dendritic cells	[271]

A summary of the effects of HDL, reconstituted and synthetic HDL, as well as HDL-associated enzymes on dendritic cell activation and functional properties is given, as described from human studies utilizing monocyte-derived dendritic cells or studies utilizing animal models. Abbreviations: apoA-I—apolipoprotein A-I; apoE—apolipoprotein E; BMDCs—bone marrow-derived dendritic cells; CD—cluster of differentiation; HDL—high-density lipoprotein; IFN-γ—interferon γ; IL—interleukin; LDL—low-density lipoprotein; MHC—major histocompatibility complex; PAF-AH—platelet-activating factor acetylhydrolase; PC—phosphatidylcholine; rHDL—reconstituted high-density lipoprotein; sHDL—synthetic high-density lipoprotein; TNF-α—tumor necrosis factor α.

**Table 6 biomedicines-09-00587-t006:** Summary of apolipoprotein mimetic peptides known to have an effect on immune cell function.

Specific Apolipoprotein Mimetic Peptides	Number of Residues	Amino Acid Sequence	References
ApoA-I mimetic peptides			
4F peptide	18	Ac-D-W-F-K-A-F-Y-D-K-V-A-E-K-F-K-E-A-F-NH_2_	[299,300]
D-4F peptide	18	Ac-D-W-F-K-A-F-Y-D-K-V-A-E-K-F-K-E-A-F-NH_2_	[301]
L-4F peptide	18	Ac-D-W-F-K-A-F-Y-D-K-V-A-E-K-F-K-E-A-F-NH_2_	[302]
5A peptide	37	D-W-L-K-A-F-Y-D-K-V-A-E-K-L-K-E-A-F-P-D-W-A-K-A-A-Y-D-K-A-A-E-K-A-K-E-A-A	[301]
ApoE mimetic peptides			
Ac-hE18A-NH_2_	28	Ac-L-R-K-L-R-K-R-L-L-R-D-W-L-K-A-F-Y-D-K-V-A-E-K-L-K-E-A-F-NH_2_	[106]

Summary of the apolipoprotein mimetic peptides known to have an effect on immune cell function. Abbreviations: apoA-I—apolipoprotein A-I; apoE—apolipoprotein.

## Data Availability

Not applicable.

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
