# Peer review of "Current Understanding of the Immunomodulatory Activities of High-Density Lipoproteins"

_biomedicines, 2021, doi:10.3390/biomedicines9060587_

Round 1

Reviewer 1 Report

This manuscript compiles the properties of HDL concerning the regulation of monocytes, macrophages, neutrophils, eosinophils, dendritic cells and T cells. The manuscript present classical knowledges and up-to date information that is relevant in the field, but here are some remarks that should be addressed to improve the manuscript.

  1. Section 3.1. Are the models of rat erythrocytes and astrocytes relevant to understand what happen with the transport of S1P in cells implicated in the coronary artery disease? The expert’s point of view is important in the interpretation of these results.
  2. Section 3.2.1. Apo CIII has been revealed as a proinflammatory peptide (Ref 73). However, the text states “…apoCII targeting might comprise a potential anti-inflammatory treatment”. It is advisable to clarify this point.
  3. Apo AII is a protein conserved along evolution, suggesting an essential role of this peptide, and the second most abundant apolipoprotein in HDL. However, this protein is only mentioned in two sentences along the manuscript. Do authors have any more information about the possible role of apo AII as immune modulator protein?
  4. Provide the region of the apolipoproteins (or the aa sequence) that corresponds to the mimetic peptides, if it is a public information of course.
  5. The ABCA1 signaling via JAK-STAT is very relevant in terms of immune response regulation and merits more emphasis in the manuscript.
  6. Section 3.6. Reference 235 describes the role of PLASMACYTOID dendritic cells. It is suggested to indicate this detail in the text, dendritic cells have different functions that plasmacytoid dendritic cells.

Author Response

Response to Reviewers

Reviewer 1

Comment 1:

This manuscript compiles the properties of HDL concerning the regulation of monocytes, macrophages, neutrophils, eosinophils, dendritic cells and T cells. The manuscript present classical knowledges and up-to date information that is relevant in the field, but here are some remarks that should be addressed to improve the manuscript.

Response 1:

We would like to thank the reviewer for the interest and the positive feedback on our manuscript.

Comment 2:

  1. Section 3.1. Are the models of rat erythrocytes and astrocytes relevant to understand what happen with the transport of S1P in cells implicated in the coronary artery disease? The expert’s point of view is important in the interpretation of these results.

Response 2:

We would like to thank the reviewer for the comment. S1P signaling has emerged as an important regulator of cardiac and vascular homeostasis and has been related to the pathogenesis of multiple cardiovascular outcomes including coronary artery disease (reviewed in detail in Diarte-Añazco E. M. G. et al. 2019 Int J Mol Sci 20(24): 6273). In addition, although erythrocytes play a fundamental role in cardiovascular homeostasis by contributing to vascular function and integrity, they also act as important triggers for the development of various cardiovascular diseases [Pernow J. et al. 2019 Cardiovascular Research 115(11): 1596-1605]. Moreover, it has been proposed by several studies that changes in microglial and astroglial activities contribute to modifications in central nervous autonomic control, and therefore, may play an important role in the development and progression of cardiovascular disease associated with increased activity of the sympathetic nervous system (Marina N. et al. 2013 Basic Res Cardiol 108, 317; Marina N. et al. 2015 Hypertension 65, 775–783; Rana I. et al. 2010 Brain Res 1326, 96-104; Shi P. et al. 2010 Hypertension 56, 297–303; Zubcevic J. et al. 2011 Hypertension 57, 1026–1033); while a recent review [Marina N. et al. 2016 Exp Physiol 101(5): 565-576] discussed in detail findings that provide novel insight into the mechanisms that link glial cell function with the pathogenesis of the cardiovascular disease. Therefore, this evidence could be of importance in coronary artery disease. However, this is not the focus of our review. To be more accurate, we added the following references in Section 3.1. [Reitsema V. et al. 2014 AIMS Mol. Sci. 1, 183–201; Diarte-Añazco E. M. G. et al. 2019 Int J Mol Sci 20(24): 6273; Nagahashi M. et al. 2013 FASEB J 27(3): 1001-1011] reviewing among others the transport of S1P from cells via specific ABC transporters, as well as the study by Lee Y. M et al. 2007 Prostaglandins Other Lipid Mediat 84(3-4): 154-162 reporting a role of ABCA1 in the release of S1P from human vascular endothelial cells.

Comment 3:

  1. Section 3.2.1. Apo CIII has been revealed as a proinflammatory peptide (Ref 73). However, the text states “…apoCII targeting might comprise a potential anti-inflammatory treatment”. It is advisable to clarify this point.

Response 3:

We would like to thank the reviewer for the comment. In the text, we state “apoC-III targeting might comprise a potential anti-inflammatory treatment” and not apoC-II as the reviewer stated above. Indeed, in the study by Zewinger S. et al. 2020 Nat Immunol 21(1): 30-41 the authors tried to identify inflammasome activators to develop new anti-inflammatory treatment strategies. Specifically, the authors identified apoC-III to activate the NLRP3 inflammasome in human monocytes, concluding therefore that targeting/inhibiting apoC-III might provide an anti-inflammatory treatment for vascular kidney diseases. To be more clear we rephrased the sentence as follows “This suggests that apoC-III inhibition might comprise a potential therapeutic target for vascular and kidney diseases”.

Comment 4:

  1. Apo AII is a protein conserved along evolution, suggesting an essential role of this peptide, and the second most abundant apolipoprotein in HDL. However, this protein is only mentioned in two sentences along the manuscript. Do authors have any more information about the possible role of apo AII as immune modulator protein?

Response 4:

We would like to thank the reviewer for the comment. As indicated by the reviewer, we added additional literature on the effects of apoA-II on immune cell function in the following sections/tables/figure:

  • Section 3.2.1. [Thompson P. A. et al. 2008 Innate Immun 14(6): 365-374].
  • Section 3.3.1. [Stein O. et al. 1995 BBA-Lipids and Lipid Metabolism 1257(2): 174-180; Wang Y. et al. 2013 Arterioscler Thromb Vasc Biol 33(2): 224-231; Rotllan N. et al. 2005 Arteriosclerosis, Thrombosis and Vascular Biology 25: E128-E132; Fournier N. et al. 2002 Arteriosclerosis, Thrombosis and Vascular Biology 22: 638-643; Melchior J. T. et al. 2017 J Lipid Res 58(7): 1374-1385].
  • Section 3.7. [Yamashita J. et al. 2011 J Immunol 186(6): 3410-3420].
  • Table 2 [Wang Y. et al. 2013 Arterioscler Thromb Vasc Biol 33(2): 224-231; Rotllan N. et al. 2005 Arteriosclerosis, Thrombosis and Vascular Biology 25: E128-E132; Fournier N. et al. 2002 Arteriosclerosis, Thrombosis and Vascular Biology 22: 638-643; Melchior J. T. et al. 2017 J Lipid Res 58(7): 1374-1385].
  • Figure 1 [Thompson P. A. et al. 2008 Innate Immun 14(6): 365-374].

Comment 5:

  1. Provide the region of the apolipoproteins (or the aa sequence) that corresponds to the mimetic peptides, if it is a public information of course.

Response 5:

As suggested by the reviewer we provide the amino acid sequence corresponding to the mimetic peptides mentioned in our revised review article in Table 6 (found in Section 4).

Comment 6:

  1. The ABCA1 signaling via JAK-STAT is very relevant in terms of immune response regulation and merits more emphasis in the manuscript.

Response 6:

As suggested by the reviewer, we have emphasized ABCA1 signaling via JAK-STAT more strongly in section 3.3.1 of our manuscript.

Comment 7:

  1. Section 3.6. Reference 235 describes the role of PLASMACYTOID dendritic cells. It is suggested to indicate this detail in the text, dendritic cells have different functions that plasmacytoid dendritic cells.

Response 7:

We would like to thank the reviewer for the comment. As suggested by the reviewer, we provide this detail in the text (Section 3.6.) as follows. “It was suggested that dendritic cells are critically involved in the progression and destabilization of atherosclerotic plaques, while in atherosclerotic plaques it was shown that plasmacytoid dendritic cells stimulate T cells against viral antigens”.

Reviewer 2 Report

Trakaki and Marsche in their review, “Current understanding of the immunomodulatory activities of high-density lipoproteins”, discuss a very comprehensive overview of HDL/HDL mimetics and its associated proteins, lipids, and enzymes. The review is very thorough and cites relevant and recent information available in this domain, and is pleasant to read. This review will be an important contribution to the field of HDL targeted therapies and I strongly recommend its acceptance for publication upon some minor changes.

I would like to point out to the authors if they can include relevant information pertaining to some additional HDL mimetics such as on "Apo A-I Milano". Many recent studies have highlighted the potential of Apo A-I Milano in treating heart failure and discussing this information would increase the coverage of the article. For example, the authors can refer to articles from Ducroux et al, Aboumsallem et al and Mishra et al.

https://www.ahajournals.org/doi/10.1161/STROKEAHA.119.027898

https://doi.org/10.1111/bph.14463

https://doi.org/10.1038/s41598-020-65255-y

https://doi.org/10.3390/biomedicines8120620

Author Response

Reviewer 2

Comment 1:

Trakaki and Marsche in their review, “Current understanding of the immunomodulatory activities of high-density lipoproteins”, discuss a very comprehensive overview of HDL/HDL mimetics and its associated proteins, lipids, and enzymes. The review is very thorough and cites relevant and recent information available in this domain, and is pleasant to read. This review will be an important contribution to the field of HDL targeted therapies and I strongly recommend its acceptance for publication upon some minor changes.

Response 1:

We would like to thank the reviewer for the interest as well as positive feedback on our review.

Comment 2:

I would like to point out to the authors if they can include relevant information pertaining to some additional HDL mimetics such as on "Apo A-I Milano". Many recent studies have highlighted the potential of Apo A-I Milano in treating heart failure and discussing this information would increase the coverage of the article. For example, the authors can refer to articles from Ducroux et al, Aboumsallem et al and Mishra et al.

https://www.ahajournals.org/doi/10.1161/STROKEAHA.119.027898

https://doi.org/10.1111/bph.14463

https://doi.org/10.1038/s41598-020-65255-y

https://doi.org/10.3390/biomedicines8120620

Response 2:

We would like to thank the reviewer for this comment. As suggested by the reviewer, we have added and discussed the relevant information in the following sections:

  • Section 3.4.3. (Ducroux C. et al. 2020 Stroke, 1886-1890)
  • Section 4 [Aboumsallem J. P. et al. 2018 BJP 175: 4167-4182; Mishra M. et al. 2020 Scientific Reports 10, 8382; Mishra M. et al. 2020 Biomedicines 8(12), 620].